**Data Availability Statement:** The datasets used in this study were obtained from the DHS program.

# Trends and correlates of low HIV knowledge among ever-married women of reproductive age: Evidence from cross-sectional Bangladesh Demographic and Health Survey 1996–2014

**Md. Tariqujjaman**[1,2,3]*, **Md. Mehedi Hasan**[1,4,5], **Mohammad Abdullah Heel Kafi**[1,6], **Md. Alamgir Hossain**[1], **Saad A. Khan**[7], **Nadia Sultana**[1,2], **Rashidul Azad**[1], **Md. Arif Hossain**[8], **Mahfuzur Rahman**[1], **Mohammad Bellal Hossain**[2]

1 International Centre for Diarrhoeal Disease Research, Dhaka, Bangladesh, 2 Department of Population Sciences, University of Dhaka, Dhaka, Bangladesh, 3 Department of Statistics, University of Dhaka, Dhaka, Bangladesh, 4 The Institute for Social Science Research, The University of Queensland, Indooroopilly, Queensland, Australia, 5 ARC Centre of Excellence for Children and Families over the Life Course (The Life Course Centre), The University of Queensland, Indooroopilly, Queensland, Australia, 6 Faculty of Medicine, The University of British Columbia, Vancouver, Canada, 7 University of Queensland, School of Biomedical Science, Queensland, Australia, 8 Department of Sociology, Jagannath University, Dhaka, Bangladesh

* md.tariqujjaman@icddrb.org

## Abstract

### Background

The human immunodeficiency virus (HIV) burden has frequently been changing over time due to epidemiological and demographic transitions. To safeguard people, particularly women of reproductive age, who can be exposed to transmitting this burden to the next generation, knowledge regarding this life-threatening virus needs to be increased. This research intends to identify the trends and associated correlates of "low" HIV knowledge among ever-married women of reproductive age in Bangladesh from 1996 to 2014.

### Methods

We analyzed data derived from six surveys of Bangladesh Demographic and Health Surveys conducted in 1996, 1999, 2004, 2007, 2011, and 2014. Analyses were primarily restricted to ever-married women aged 15–49 years who had ever heard of HIV. The correlates of "low" HIV knowledge were investigated using multiple binary logistic regression models.

### Results

The study found that the proportion of women with "low" HIV knowledge decreased from 72% in 1996 to 58% in 2014. In adjusted models, age at first marriage, level of education, wealth quintile, and place of residence (except in the survey year 2011) were found to be potential correlates of "low" HIV knowledge in all survey years. In the pooled analysis, we

All the data were downloaded from DHS website (https://dhsprogram.com/data/available-datasets.cfm) after authorization was received on the data request. Since the data set is publicly available, contingent upon getting authorization from DHS Program website, we cannot upload the data set here.

**Funding:** The author(s) received no specific funding for this work.

**Competing interests:** The authors have declared that no competing interests exist.

found lower odds of "low" HIV knowledge in the survey years 1999 (Adjusted Odds Ratio: 0.67; 95% CI: 0.57, 0.78), 2004 (AOR: 0.60; 95% CI: 0.52, 0.70), 2007 (AOR: 0.51; 95% CI: 0.44, 0.60), 2011 (AOR: 0.36; 95% CI: 0.32, 0.42) and 2014 (AOR: 0.47; 95% CI: 0.41, 0.54) compared to the survey year 1996.

## Conclusion

The proportion of "low" HIV knowledge has declined over time, although the proportion of women with "low" HIV knowledge still remains high. The prevention of early marriage, the inclusion of HIV-related topics in the curricula, reduction of disparities between urban-rural and the poorest-richest groups may help to improve the level of HIV knowledge among ever-married Bangladeshi women.

## Introduction

The human immunodeficiency virus (HIV) is one of the leading public health challenges globally, especially in low- and middle-income countries (LMICs) [1, 2]. Globally, in 2020, 680,000 people died from HIV-related causes, and 1.5 million had been infected with HIV [3]. There are 18.8 million women aged 15 or above living with HIV, worldwide [4]. Africa is a region where the HIV infection rate is remarkably high, 37.7 million people are currently living with HIV [3]. Approximately 3.5 million people are living with HIV in Southeast Asia [5]. Globally, it is predicted that 47% of new HIV infections will be found among the most at-risk populations, such as gay men, injecting drug users, the prison population, male and female sex workers, and transgender individuals [3, 6].

Although the prevalence of HIV among the general population in Bangladesh is low (0.1%), there is a higher risk of infection and transmission among the most at-risk groups mentioned above as well as people living in broader areas, floating people, and bridging populations [3, 6–8]. Moreover, since 2017, there has been an influx of approximately half a million Rohingya (an ethnic group who mostly follow Islam religion and resided in Rakhine State) from Myanmar to Bangladesh and taken shelter in Cox's Bazar district. In Rohingya camps, more than 5000 individuals are found HIV positive [9].

In Bangladesh, the prevalence of HIV among women is below 0.1% [10, 11]. However, women are more prone to unintentionally spread HIV infections compared to men because of the possibility of HIV transmission during sexual intercourse, from mother to child during pregnancy, and breastfeeding [3, 12, 13]. Moreover, social stigma, lack of freedom of choice of condom use and empowerment, fear of violence, and discrimination are all additional factors contributing to women's increased vulnerability to HIV infection [14]. Furthermore, women are at an increased risk of contracting an HIV infection due to the lack of HIV testing and counseling, low treatment adherence, and low prevalence of condom usage during sexual intercourse [12]. Behavioral factors such as sexual exposure at an early age and having sex with multiple partners, especially among the bridging population, can also increase the risk of HIV infection among women [15]. In addition, many misconceptions and rumors regarding HIV, such as people getting HIV from mosquito bites, sharing food with a person who has AIDS, and by witchcraft or supernatural means exist among ever-married women [16]. Therefore, correct knowledge of HIV, especially among married women who are at the reproductive stage, is essential to mitigating this transmission. As such, it is also necessary to identify

potential correlates of knowledge regarding HIV as this will help in the adoption of more effec-
tive long-term policy objectives and interventions.

In Bangladesh, several studies have explored the correlates of HIV but have not demon-
strated changing trends of HIV prevalence, particularly for women of reproductive age [17–
21]. One study looked at the trends and determinants of HIV knowledge and awareness from
2004 to 2014 and considered whether participants had ever heard of HIV as a knowledge and
awareness outcome measure [22]. However, we believe that only ever heard of HIV does not
fully reflect the actual knowledge about HIV. Rather scoring from correct responses related to
specific HIV knowledge would better reflect women's knowledge about HIV. In this study, we
aimed to discover trends, as well as the changes in the correlates of "low" HIV knowledge
among ever-married Bangladeshi women of reproductive age who have ever heard of HIV.

## Methodology

### Data source

Data for this study were derived from the Bangladesh Demographic and Health Surveys
(BDHS) conducted in the years 1996 (n = 9127), 1999 (n = 10544), 2004 (n = 11440), 2007
(n = 10996), 2011 (n = 17842), and 2014 (n = 17863). We analyzed the sample of ever-married
women of reproductive age (15–49 years) who ever heard of HIV. Therefore, our sample sizes
for this study were 1781, 3660, 7235, 7687, 12512, and 12593 in the survey years 1996, 1999,
2004, 2007, 2011, and 2014, respectively (Fig 1) [23]. We did not include the latest BDHS data
(2017–2018) because, in this round of the survey, the HIV data were not collected. The cross-
sectional surveys were conducted in collaboration with the National Institute of Population
Research and Training (NIPORT), the Inner City Fund (ICF) International USA, and Mitra &

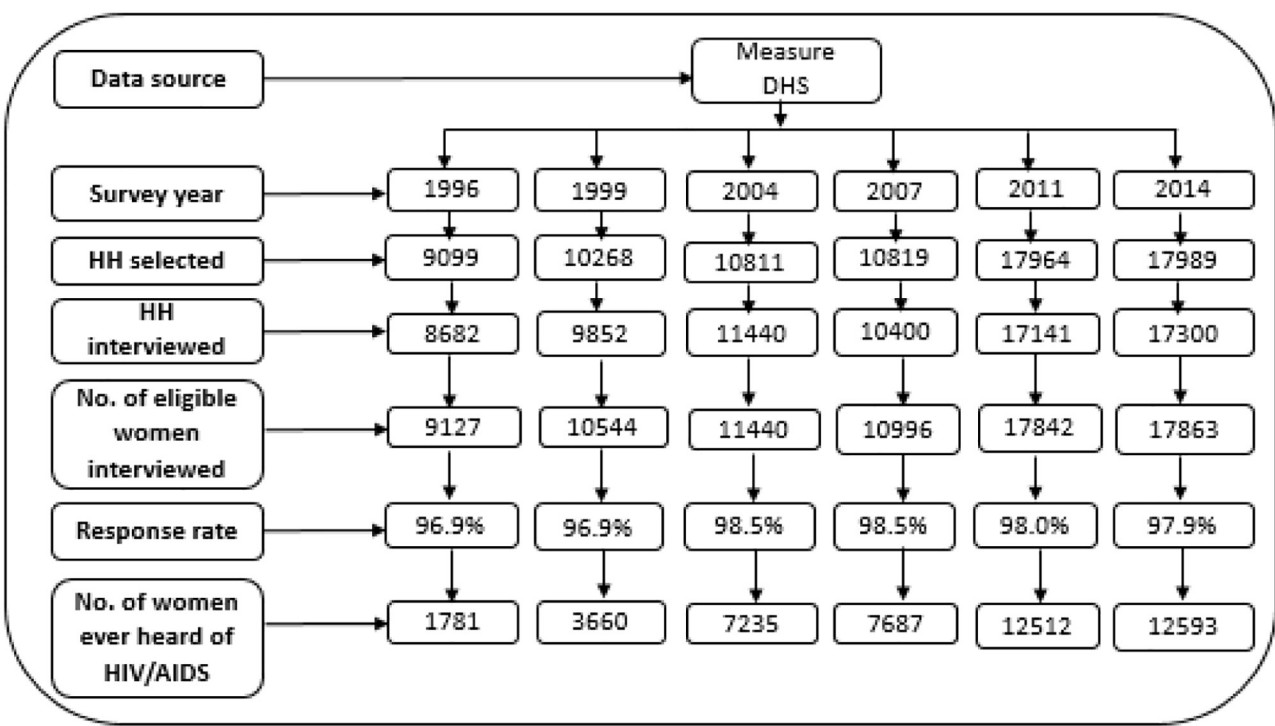

**Fig 1. Flow-chart of data extraction for analysis in this study.**

Associates. The BDHS compiles information on a variety of sociodemographic and health-related indicators including the socio-economic status of households, fertility, and reproductive health, maternal and newborn health, nutritional status of women and children, women empowerment, healthcare-seeking behavior, and knowledge, attitudes, and behaviors regarding HIV and other sexually transmitted infections.

## Sampling design

The BDHS employed a two-stage stratified cluster sampling technique to survey respondent households. The sampling frame was a complete list of enumeration areas or clusters, consisting of either a village, part of a village, or a group of villages. In the first stage of sampling, 313 clusters (71 urban and 242 rural) were selected in 1996, 341 clusters (99 urban and 242 rural) were selected in 1999, 361 clusters (122 urban and 239 rural) were selected in 2004, 364 clusters (136 urban and 228 rural) were selected in 2007, 600 clusters (207 urban and 393 rural) were selected in 2011 and 619 clusters (226 urban and 393 rural) were selected in 2014, using probability proportional to size sampling. In the second stage of sampling, 30 households were selected using the systematic random sampling technique. Overall, the sampling technique used in the BDHS was designed to be representative of the Bangladeshi population, and the selected clusters and households were chosen based on a rigorous statistical methodology.

## Outcome measure

The outcome variable of this study was HIV-related knowledge. A set of yes/no/don't know type questions was administered to gather information regarding HIV knowledge from the respondents in different survey years. The correct answer to each question was coded as "1", and the incorrect answers were coded as "0". Then, the score of all questions of each respondent was summed. Finally, the knowledge score was categorized as either a "low score" or a "high score" based on the median value of the summed scores of the total questions (median or below "low score" and above the median "high score") [24]. We operationally define the "high score" as "high knowledge" and "low score" as "low knowledge". Across the surveys, all the HIV-related knowledge questions were not uniform because the BDHS questionnaire is upgradable depending on the new knowledge, indicators, aspects, and issues generated/identified from up-to-date research findings. The internal reliability (Cronbach Alpha) of the set of knowledge questions at each survey year was measured: 0.68 in 1996 (12 questions), 0.59 in 1999 (14 questions), 0.77 in 2004 (14 questions), 0.59 in 2007 (8 questions), 0.61 in 2011 (11 questions) and 0.63 in 2014 (11 questions). All the questions that were recorded to generate knowledge scores are presented in Table 1.

## Covariate measures

We conducted an extensive literature review to select the relevant covariates of knowledge about HIV [17, 22, 25–28]. In this study, a set of covariates was included, including respondent's age (categorized as 15–19 years, 20–29 years, 30–39 years, and 40–49 years), age at first marriage (categorized as <15 years, 15–17 years and ≥18 years), respondent's level of education (categorized as No formal education, Primary, Secondary, and Higher), religion (categorized as Islam and Others—Hinduism, Buddhism, and Christianity), sex of the household head (Male, Female), respondent's current marital status (categorized as Married and Others—widowed, divorced and not living together), type of place of residence (Urban, Rural), respondent's current employment status (Unemployed, Employed), use of condom during sexual intercourse (No, Yes). Women's exposure to mass media was characterized in terms of reading newspapers, listening to the radio, or watching television at least once a week. The

**Table 1. List of questions with original codes and labels and converted codes and labels to create binary variables for constructing knowledge scores.**

| Questions related to HIV/AIDS knowledge | Original coded value | Original label | Recoded value | Revised Label |
|---|---|---|---|---|
| People can reduce their chance of getting the AIDS virus by having just one uninfected sex partner, who has no other sex partners | 0 | No | 0 | No knowledge |
| | 1 | Yes | 1 | Have knowledge |
| | 8 | Don't know | 0 | No knowledge |
| People can reduce their chance of getting the AIDS virus by using a condom every time they have sex | 0 | No | 0 | No knowledge |
| | 1 | Yes | 1 | Have knowledge |
| | 8 | Don't know | 0 | No knowledge |
| People get AIDS virus by unsafe blood transfusion | 0 | No | 0 | No knowledge |
| | 1 | Yes | 1 | Have knowledge |
| | 8 | Don't know | 0 | No knowledge |
| People get the AIDS virus by using unsterilized needle or syringe | 0 | No | 0 | No knowledge |
| | 1 | Yes | 1 | Have knowledge |
| | 8 | Don't know | 0 | No knowledge |
| People can get the AIDS virus because of witchcraft or other supernatural means | 0 | No | 1 | Have knowledge |
| | 1 | Yes | 0 | No knowledge |
| | 8 | Don't know | 0 | No knowledge |
| AIDS transmitted from mother to her baby by breastfeeding | 0 | No | 0 | No knowledge |
| | 1 | Yes | 1 | Have knowledge |
| | 8 | Don't know | 0 | No knowledge |
| AIDS transmitted from mother to her baby during delivery | 0 | No | 0 | No knowledge |
| | 1 | Yes | 1 | Have knowledge |
| | 8 | Don't know | 0 | No knowledge |
| AIDS transmitted from mother to her baby during pregnancy | 0 | No | 0 | No knowledge |
| | 1 | Yes | 1 | Have knowledge |
| | 8 | Don't know | 0 | No knowledge |
| It is possible a healthy-looking person can have AIDS virus | 0 | No | 0 | No knowledge |
| | 1 | Yes | 1 | Have knowledge |
| | 8 | Don't know | 0 | No knowledge |
| People get AIDS by sharing food with a person who has AIDS | 0 | No | 1 | Have knowledge |
| | 1 | Yes | 0 | No knowledge |
| | 8 | Don't know | 0 | No knowledge |
| People get the AIDS virus from mosquito bites | 0 | No | 1 | Have knowledge |
| | 1 | Yes | 0 | No knowledge |
| | 8 | Don't know | 0 | No knowledge |
| There are ways to avoid AIDS | 0 | No | 0 | No knowledge |
| | 1 | Yes | 1 | Have knowledge |
| | 8 | Don't know | 0 | No knowledge |

(*Continued*)

**Table 1.** (Continued)

| Questions related to HIV/AIDS knowledge | Original coded value | Original label | Recoded value | Revised Label |
|---|---|---|---|---|
| AIDS can avoid by abstaining from sex | 0 | No | 0 | No knowledge |
| | 1 | Yes | 1 | Have knowledge |
| | 8 | Don't know | 0 | No knowledge |
| AIDS can avoid by avoiding sex with prostitute | 0 | No | 0 | No knowledge |
| | 1 | Yes | 1 | Have knowledge |
| | 8 | Don't know | 0 | No knowledge |
| AIDS can spread by kissing | 0 | No | 1 | Have knowledge |
| | 1 | Yes | 0 | No knowledge |
| | 8 | Don't know | 0 | No knowledge |
| People avoid AIDS by Traditional healer | 0 | No | 1 | Have knowledge |
| | 1 | Yes | 0 | No knowledge |
| | 8 | Don't know | 0 | No knowledge |
| AIDS is a fatal disease | 0 | No | 0 | No knowledge |
| | 1 | Yes | 1 | Have knowledge |
| | 8 | Don't know | 0 | No knowledge |
| AIDS can avoid by avoiding sex with homosexual | 0 | No | 0 | No knowledge |
| | 1 | Yes | 1 | Have knowledge |
| | 8 | Don't know | 0 | No knowledge |
| AIDS can avoid by limiting sex with marriage or avoid sex with many partners | 0 | No | 0 | No knowledge |
| | 1 | Yes | 1 | Have knowledge |
| | 8 | Don't know | 0 | No knowledge |
| AIDS can avoid by avoiding sex with intravenous drug users | 0 | No | 0 | No knowledge |
| | 1 | Yes | 1 | Have knowledge |
| | 8 | Don't know | 0 | No knowledge |

respondents were categorized as 'yes' if they were exposed to at least one type of media at least once a week; otherwise, they were categorized as 'no'. We also considered administrative divisions (Barisal, Chattogram, Dhaka, Khulna, Rajshahi, Rangpur, Sylhet) and wealth quintile (Poorest, Poorer, Middle, Richer, Richest) as potential covariates of this study. However, the variable wealth quintile was not in existence in the 1996 and 1999 surveys. Therefore, we calculated the wealth quintile using principal component analysis in these two survey years based on the ownership of selected assets, household structure (materials used for floor, roof, and wall of the house), type of latrine installed, and sources of drinking water in the same way the Demographic and Health Survey (DHS) constructed in the other survey years [29]. Rangpur was not an administrative division before 2011 in Bangladesh. This division was part of the Rajshahi division in earlier surveys. We considered Rangpur as a distinct administrative division in the survey years 2011 and 2014. We also considered survey years (1996, 1999, 2004, 2007, 2011, and 2014) as a potential covariate for multiple regression in the pooled analysis to control the variations due to different survey times.

## Statistical analysis

Univariate analysis was performed and presented the estimates in frequencies and percentages along with their respective 95% confidence intervals (CIs) where necessary. All univariate analyses were conducted by considering the complex survey design for capturing variations due to weighting and study design. Bivariate analysis was carried out using a simple logistic regression model to measure the association between "low" HIV knowledge and different covariates. The results were presented as unadjusted odds ratios with 95% CIs. Finally, multiple binary logistic regression analysis was performed to explore the correlates of "low" HIV knowledge in different survey years separately, as well as for the pooled data. The estimates were presented in adjusted odds ratios (AORs) with 95% CIs. In the multiple binary logistic regression model, we entered only those covariates that were found to be significant (p-value <0.05) in the simple logistic regression models. Cluster (primary sampling units) variations were adjusted while performing regression analyses by using the "cluster" command in Stata. We also conducted a sensitivity analysis that included both samples of ever-married women who ever heard of HIV and those who never heard of it. The never heard of HIV was classified as an incorrect answer for each of the knowledge questions. We constructed the outcome variable that considered "never heard of HIV" as "no" knowledge. We categorized the outcome variable for sensitivity analysis in the same way as our main analysis. The goodness of model fitting was checked by the Hosmer Lemeshow test. The total variations of covariates were expressed using the area under the curve. All analyses were performed using the statistical software package Stata, version 15.0 SE (StataCorp. LP, College Station, TX, USA).

## Ethics statements

The BDHS was conducted under the authority of the NIPORT of the Ministry of Health and Family Planning. Mitra and Associates, a Bangladeshi research firm, implemented the survey. The ICF International provided technical assistance to the survey as part of its Demographic and Health Survey Programs (MEASURE DHS). The survey methodology and questionnaire were reviewed and approved by the Institutional Review Board of ICF. The BDHS obtained written consent from the respondents before conducting the interviews.

## Results

### Sample characteristics

The mean age of the women was about 30 years, and the highest percentage of women belonged to the 20–29 years age group (range: 41.1%—43.2%) in all surveys. The mean age of first marriage was about 16 years, and the majority (range: 32.1%—48.3%) of women first married before the age of 15 years in all the surveys. The number of women with no formal education decreased from 19% in 1996 to 15% in 2014. The percentage of women from urban communities ranged between 29% and 36% throughout the survey years. The use of condoms during the sexual intercourse was also relatively low (range: 6%—11%) (Table 2).

### Trends of HIV-related knowledge from 1996 to 2014

The trends of HIV knowledge among ever-married women of reproductive age are presented in Fig 2. In 1996, 72% of women had low knowledge concerning HIV; it decreased to 66% in 1999, and then slightly increased to 66.3% in 2004. Further, the percentage of low knowledge was 62.6% in 2007. It decreased to 53% in 2011 and further increased to 57.6% in 2014. The pooled estimate of low HIV knowledge was 60% among ever-married women.

**Table 2. Background characteristics of the study participants, 1996 to 2014.**

| Characteristics | 1996 (n = 1781) | 1999 (n = 3660) | 2004 (n = 7235) | 2007 (n = 7687) | 2011 (n = 12512) | 2014 (n = 12593) | Pooled Data (n = 45468) |
|---|---|---|---|---|---|---|---|
| **Age of respondent** | | | | | | | |
| Mean (SD) | 29.1 (8.5) | 29.2 (8.8) | 28.8 (9.1) | 29.5 (9.0) | 29.8 (8.9) | 30.2 (8.9) | 29.6 (8.9) |
| **Age categories, % (n)** | | | | | | | |
| 15–19 years | 14.6 (243) | 15.3 (527) | 17.9 (1232) | 14.8 (1051) | 12.1 (1441) | 11.8 (1462) | 13.7 (5956) |
| 20–29 years | 42.4 (760) | 41.9 (1509) | 40.5 (2910) | 41.6 (3173) | 43.2 (5286) | 41.1 (5066) | 41.8 (18704) |
| 30–39 years | 28.3 (513) | 27.6 (1049) | 26.3 (1943) | 27.9 (2196) | 26.9 (3436) | 29.3 (3699) | 28.3 (12836) |
| 40–49 years | 14.7 (265) | 15.2 (575) | 15.3 (1150) | 15.8 (1267) | 17.9 (2293) | 17.8 (2366) | 17.4 (7916) |
| **Age at first marriage** | | | | | | | |
| Mean (SD) | 15.8 (3.8) | 16.2 (3.5) | 15.4 (3.0) | 15.8 (3.0) | 16.1 (3.1) | 16.2 (3.1) | 15.9 (3.1) |
| **Categories, % (n)** | | | | | | | |
| <15 years | 42.4 (744) | 38.3 (1345) | 48.3 (3369) | 41.2 (2936) | 36.0 (4374) | 32.1 (3968) | 38.1 (16736) |
| 15–17 years | 31.0 (545) | 34.91280 | 33.9 (2470) | 38.4 (2952) | 39.0 (4870) | 40.8 (5138) | 38.0 (17255) |
| ≥ 18 years | 26.7 (492) | 26.91035 | 17.7 (1396) | 20.3 (1799) | 25.0 (3268) | 27.2 (3487) | 25.2 (11477) |
| **Respondent's education, % (n)** | | | | | | | |
| No formal education | 18.6 (328) | 18.4 (630) | 25.5 (1739) | 21.5 (1579) | 16.1 (1880) | 14.5 (1713) | 18.3 (7869) |
| Primary | 24.5 (431) | 25.0 (889) | 30.7 (2190) | 28.8 (2197) | 27.9 (3418) | 26.6 (3320) | 27.8 (12445) |
| Secondary or higher | 56.9 (1022) | 56.5 (2141) | 43.8 (3306) | 49.8 (3911) | 56.0 (7214) | 49.0 (7560) | 54.0 (25154) |
| **Religion, % (n)** | | | | | | | |
| Others[a] | 12.3 (231) | 13.7 (565) | 10.9 (845) | 8.8 (734) | 9.9 (1388) | 9.1 (1139) | 10.0 (4902) |
| Islam | 87.7 (1550) | 86.3 (3095) | 89.1 (6390) | 91.2 (6953) | 90.1(11124) | 91.9 (11454) | 90.0 (40566) |
| **Current employment status, % (n)** | | | | | | | |
| Unemployed | 69.4 (1251) | 78.9 (2926) | 79.0 (5746) | 70.2 (5568) | 86.2(10779) | 68.5 (8783) | 76.1 (35053) |
| Employed | 30.6 (530) | 21.1 (734) | 21.0 (1489) | 29.8 (2119) | 13.9 (1733) | 31.5 (3810) | 23.9 (10415) |
| **Current marital status, % (n)** | | | | | | | |
| Others[b] | 5.4 (97) | 5.8 (207) | 6.0 (457) | 6.0 (484) | 5.2 (662) | 4.4 (593) | 5.3 (2500) |
| Married | 94.6 (16.8) | 94.2 (3253) | 94.0 (6778) | 94.0 (7203) | 94.8(11850) | 95.6 (12000) | 94.7 (42968) |
| **Media exposure, % (n)** | | | | | | | |
| No | 14.1 (240) | 17.6 (577) | 13.8 (956) | 51.0 (4020) | 22.7 (2738) | 24.6 (3094) | 25.9 (11625) |
| Yes | 85.9 (1541) | 82.4 (3083) | 86.3 (6279) | 49.0 (3667) | 77.3 (9774) | 75.4 (9499) | 74.1 (33843) |
| **Use of condom during sexual intercourse, % (n)** | | | | | | | |
| No | 89.3 (1584) | 91.4 (3317) | 94.3 (6778) | 94.3 (7177) | 93.1(11574) | 92.0 (11567) | 92.9 (41997) |
| Yes | 10.8 (197) | 8.6 (343) | 5.7 (457) | 5.8 (510) | 6.9 (938) | 8.0 (1026) | 7.1 (3471) |
| **Sex of the household head, % (n)** | | | | | | | |
| Male | 91.3 (1626) | 90.9 (3348) | 90.4 (6560) | 87.9 (6782) | 89.5(11203) | 88.6 (11082) | 89.3 (40601) |
| Female | 8.7 (155) | 9.1 (312) | 9.6 (675) | 12.1 (905) | 10.5 (1309) | 11.4 (1511) | 10.7 (4867) |
| **Type of place of residence, % (n)** | | | | | | | |
| Urban | 36.4 (749) | 41.1 (1974) | 30.8 (3188) | 29.2 (3549) | 32.2 (5182) | 34.5 (5125) | 32.9 (19767) |
| Rural | 63.7 (1032) | 58.9 (1686) | 69.2 (4047) | 70.8 (4138) | 67.8 (7330) | 65.5 (7468) | 67.1 (25701) |
| **Wealth quintile, % (n)** | | | | | | | |
| Poorest | 6.6 (114) | 10.7 (359) | 9.6 (614) | 11.4 (725) | 11.4 (1324) | 11.6 (1413) | 11.0 (4549) |
| Poorer | 8.1 (142) | 15.8 (535) | 14.9 (935) | 15.4 (1073) | 15.2 (1805) | 15.2 (1900) | 15.0 (6390) |
| Middle | 9.3 (163) | 12.2 (375) | 19.4 (1275) | 19.6 (1382) | 20.3 (2408) | 20.5 (2600) | 19.1 (8203) |
| Richer | 21.1 (348) | 24.7 (822) | 25.2 (1720) | 25.2 (1819) | 24.3 (3062) | 24.5 (3083) | 24.5 (10854) |
| Richest | 54.8 (1014) | 36.7 (1562) | 30.9 (2691) | 28.4 (2688) | 28.7 (3913) | 28.2 (3597) | 30.5 (15465) |
| **Administrative division, % (n)** | | | | | | | |
| Barisal | 6.8 (185) | 6.6 (344) | 6.3 (880) | 5.5 (957) | 5.8 (1521) | 6.4 (1570) | 6.1 (5457) |
| Chattogram | 21.8 (299) | 19.4 (697) | 17.6 (1292) | 17.8 (1311) | 18.0 (2028) | 18.3 (2041) | 18.3 (7668) |

*(Continued)*

**Table 2.** (Continued)

| Characteristics | 1996 (n = 1781) | 1999 (n = 3660) | 2004 (n = 7235) | 2007 (n = 7687) | 2011 (n = 12512) | 2014 (n = 12593) | Pooled Data (n = 45468) |
|---|---|---|---|---|---|---|---|
| Dhaka | 42.8 (652) | 38.0 (1091) | 34.8 (1802) | 33.9 (1795) | 35.1 (2335) | 37.1 (2403) | 35.9 (10078) |
| Khulna | 11.5 (217) | 13.7 (689) | 14.8 (1284) | 14.7 (1375) | 13.8 (21.6) | 11.4 (2035) | 13.3 (7757) |
| Rajshahi | 12.7 (279) | 18.4 (541) | 21.8 (1388) | 22.9 (1364) | 13.6 (1727) | 10.7 (1669) | 15.9 (6968) |
| Rangpur | - | - | - | - | 9.1 (1445) | 10.3 (1556) | 3.0 (2618) |
| Sylhet | 4.4 (149) | 3.8 (298) | 4.8 (589) | 5.3 (885) | 4.6 (1299) | 5.9 (1319) | 7.5 (4922) |

[a]Hinduism, Buddhism, and Christianity
[b]Widowed /Divorced/Not living together

In both urban and rural areas, we found decreasing trends of low knowledge. But there were significant differences in low knowledge among women living in urban and rural areas in all the survey years (urban area: range 46.8%—59.7%, rural area: range 56%—78.2%). We observed differences in low HIV knowledge among women who lived in the poorest and richest wealth quintiles in all the survey years (poorest: range 63.7%—90%, richest: 42%—60.5%) (Fig 3). The distribution of individual knowledge responses was presented in **S1 Table (S1 File)**.

## Correlates of "low" HIV knowledge from 1996 to 2014

We explored significant correlates of low HIV knowledge with the respondents' background characteristics using a simple logistic regression model (S2 Table in S1 File). We entered covariates for multiple models, which were significant (p<0.05) in a simple logistic regression

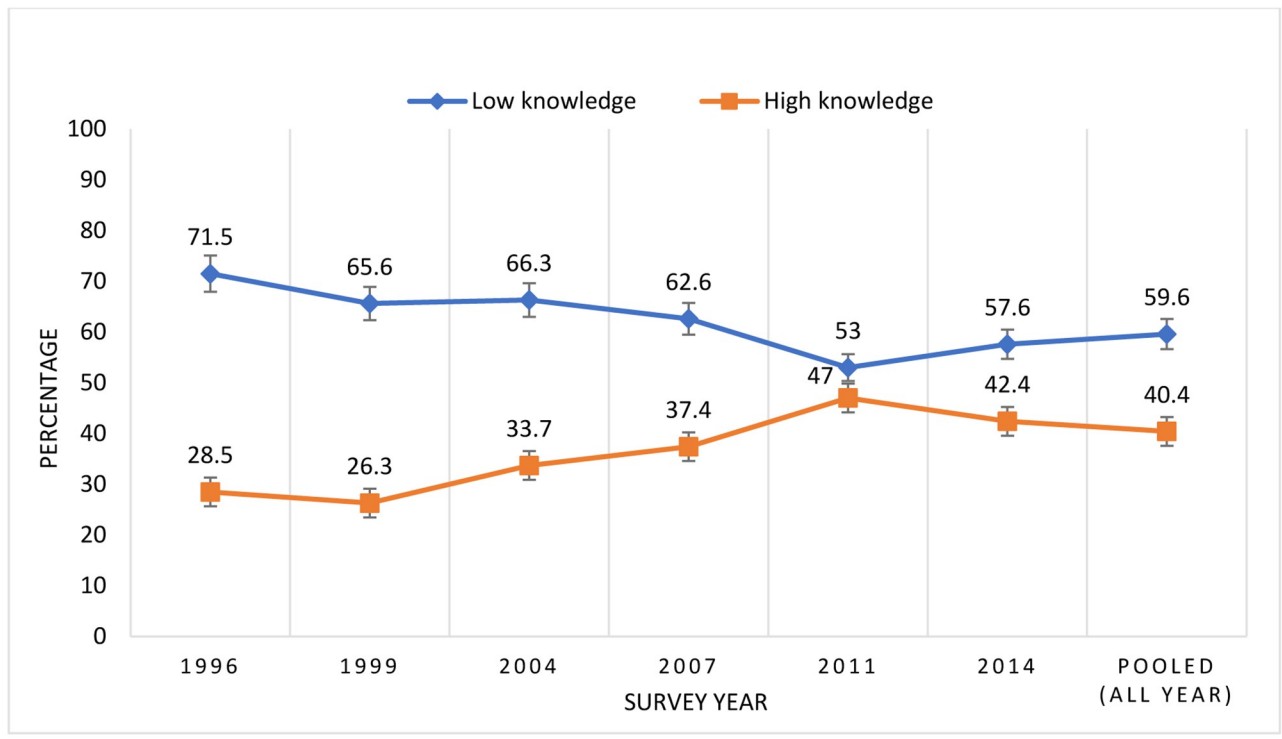

**Fig 2. Trends of HIV knowledge who ever heard of HIV from 1996 to 2014.**

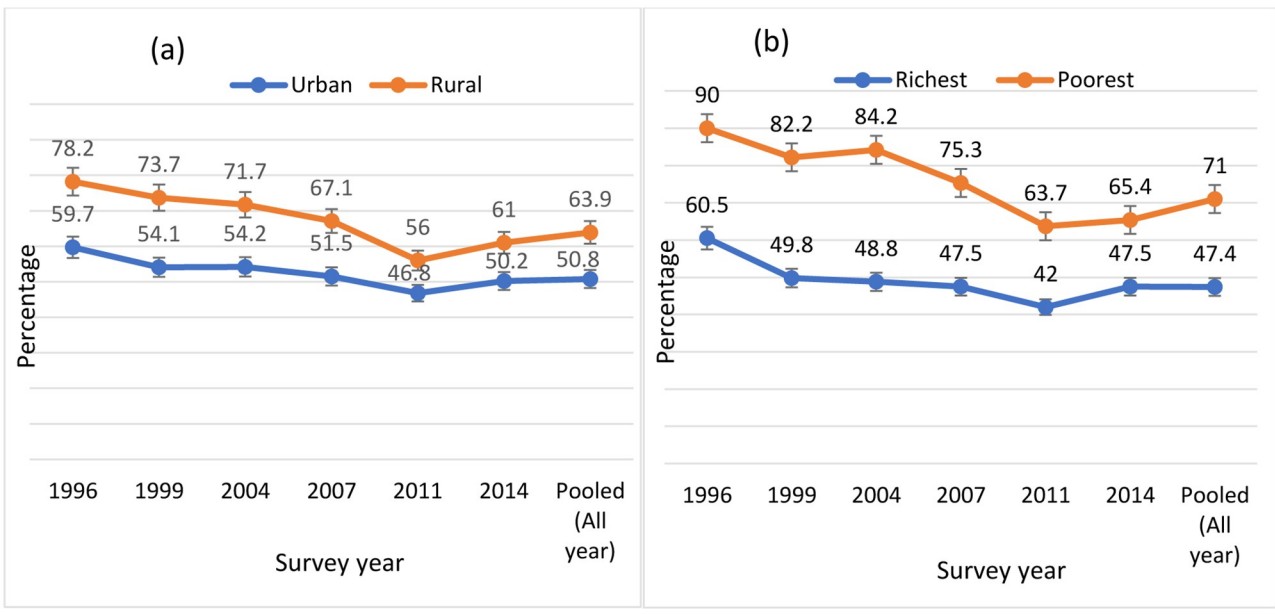

**Fig 3. Urban-rural and richest-poorest differentials of low HIV knowledge.**

model. The results of multiple logistic regression models of low HIV knowledge among ever-married women of reproductive age from 1996 to 2014 and pooled data are presented in Table 3. In our pooled data, we explored lower odds of "low" HIV knowledge among women with age of first marriage 18 and above (AOR: 0.75; 95% CI: 0.71, 0.80) than women with age at first marriage below 15 years. Conversely, women with no formal education (AOR: 2.32; 95% CI: 2.17, 2.49), women with no media exposure (AOR: 1.12; 95% CI: 1.05, 1.28), women lived in rural areas (AOR: 1.32; 95% CI: 1.24, 1.40) and belonged to the poorest (AOR: 1.70; 95% CI: 1.55, 1.87 wealth quintile had higher odds of "low" HIV knowledge than their counterpart categories. In the pooled data, we found lower odds of having a low HIV knowledge throughout the survey period compared to the survey year 1996 (AOR range: 0.36–0.67).

In individual survey data, we found women who first married at the age of eighteen and above had less likely (AOR range: 0.68–0.80) of "low" HIV knowledge in all survey years than women first married before the age of fifteen. Women with no formal education had higher odds (AOR range: 1.82–5.64) of having a "low" HIV knowledge than women who completed secondary or higher education in all survey years. Adjusted models further revealed that women who were not exposed to media had a higher likelihood (except in 2007 and 2014) of "low" HIV knowledge than women who were exposed to any media (AOR range:1.12–2.93). In all survey years (except in the survey year 2011), we found significantly higher odds (AOR range:1.28–1.77) of "low" HIV knowledge among rural women, indicating women residing in rural areas had a higher likelihood of "low" HIV knowledge than their urban counterparts. Compared to women living in the richest quintile, women living in the poorer and poorest quintile had higher odds of "low" HIV knowledge (AOR range: 1.26–2.79). Additionally, unemployed women were 1.53 times more likely (in the survey year 1999) and 1.16 times more likely (in the survey year 2011) to have a "low" HIV knowledge than employed women. In the case of the division of Bangladesh, women living in the Dhaka division in the survey year 2011 had a significantly higher likelihood (AOR: 1.41; 95% CI: 1.17, 1.69) of "low" HIV knowledge compared to those women residing in the Barisal division. On the other hand,

**Table 3. Correlates of "low" HIV knowledge among ever-married women from 1996 to 2014 (Multiple logistic regression model).**

| Characteristics | 1996 AOR (95% CI) | 1999 AOR (95% CI) | 2004 AOR (95% CI) | 2007 AOR (95% CI) | 2011 AOR (95% CI) | 2014 AOR (95% CI) | Pooled data AOR (95% CI) |
|---|---|---|---|---|---|---|---|
| **Age categories** | | | | | | | |
| 15–19 years | Ref. | Ref. | Ref. | Ref. | Ref. | Ref. | Ref. |
| 20–29 years | 0.63* (0.43, 0.93) | 0.65*** (0.52, 0.82) | 0.86* (0.75, 0.99) | 1.09 (0.86, 1.18) | 0.85** (0.75, 0.96) | 0.99 (0.87, 1.12) | 0.88*** (0.83, 0.94) |
| 30–39 years | 0.53** (0.36, 0.77) | 0.74* (0.59, 0.95) | 0.97 (0.82, 1.15) | 1.03 (0.88, 1.22) | 0.78*** (0.68, 0.89) | 0.93 (0.82, 1.06) | 0.86*** (0.80, 0.92) |
| 40–49 years | 0.54** (0.34, 0.85) | 0.91 (0.67, 1.23) | 1.21 (0.99, 1.48) | 1.11 (0.91, 1.35) | 0.82** (0.70, 0.96) | 0.97 (0.83, 1.23) | 0.93 (0.85, 1.00) |
| **Age at first marriage** | | | | | | | |
| <15 years | Ref. | Ref. | Ref. | Ref. | Ref. | Ref. | Ref. |
| 15–17 years | 0.83 (0.62, 1.10) | 1.09 (0.90, 1.31) | 0.84** (0.75, 0.95) | 1.00 (0.90, 1.12) | 0.88** (0.81, 0.96) | 0.91* (0.83, 0.99) | 0.90*** (0.86, 0.94) |
| ≥18 years | 0.68** (0.51, 0.89) | 0.80* (0.64, 0.99) | 0.73*** (0.63, 0.85) | 0.76*** (0.66, 0.87) | 0.78*** (0.71, 0.87) | 0.79*** (0.71, 0.88) | 0.75*** (0.71, 0.80) |
| **Education level** | | | | | | | |
| No formal education | 5.64*** (3.62, 8.80) | 2.7*** (2.12, 3.60) | 2.64*** (2.23, 3.14) | 2.54*** (2.15, 3.01) | 1.99*** (1.74, 2.27) | 1.82*** (1.59, 2.10) | 2.32*** (2.17, 2.49) |
| Primary | 2.94*** (2.13, 4.05) | 2.31*** (1.86, 2.87) | 1.81*** (1.59, 2.07) | 1.87*** (1.65, 2.12) | 1.53*** (1.39, 1.69) | 1.61*** (1.46, 1.78) | 1.76*** (1.67, 1.85) |
| Secondary or higher | Ref. | Ref. | Ref. | Ref. | Ref. | Ref. | Ref. |
| **Employment** | | | | | | | |
| Unemployed | - | 1.53*** (1.27, 1.86) | - | - | 1.16* (1.03, 1.30) | - | - |
| Employed | | Ref. | | | Ref. | | |
| **Current marital status** | | | | | | | |
| Married | | Ref. | Ref. | Ref. | Ref. | Ref. | Ref. |
| Other¶ | - | 1.66** (1.15, 2.40) | 1.17 (0.91, 1.49) | 1.17 (0.94, 1.47) | 1.17 (0.99, 1.38) | 1.08 (0.89, 1.31) | 1.16** (1.05, 1.28) |
| **Media exposure** | | | | | | | |
| No | 2.26** (1.35, 3.79) | 1.65** (1.24, 2.21) | 2.93*** (2.32, 3.70) | - | 1.12* (1.00, 1.25) | - | 1.12*** (1.05, 1.18) |
| Yes | Ref. | Ref. | Ref. | | Ref. | | Ref. |
| **Use of condom** | | | | | | | |
| No | 1.26 (0.92, 1.74) | 1.85*** (1.48, 2.32) | 1.63*** (1.30, 2.05) | 1.88*** (1.54, 2.30) | 1.34*** (1.16, 1.54) | 1.24** (1.09, 1.42) | 1.43*** (1.33, 1.54) |
| Yes | Ref. | Ref. | Ref. | Ref. | Ref. | Ref. | Ref. |
| **Type of place of residence** | | | | | | | |
| Urban | Ref. | Ref. | Ref. | Ref. | Ref. | Ref. | Ref. |
| Rural | 1.54** (1.18, 2.01) | 1.77*** (1.46, 2.15) | 1.51*** (1.28, 1.77) | 1.40*** (1.20, 1.63) | 1.08 (0.97, 1.20) | 1.28*** (1.14, 1.43) | 1.32*** (1.24, 1.40) |
| **Wealth quintile** | | | | | | | |
| Richest | Ref. | Ref. | Ref. | Ref. | Ref. | Ref. | Ref. |
| Richer | 1.66** (1.21, 2.27) | 1.50*** (1.21, 1.87) | 1.39*** (1.19, 1.61) | 1.24** (1.06, 1.44) | 1.29*** (1.15, 1.44) | 1.14* (1.02, 1.29) | 1.31*** (1.23, 1.39) |
| Middle | 1.96* (1.12, 3.43) | 1.78 (1.34, 2.38) | 1.90*** (1.59, 2.27) | 1.68 (1.40, 2.00) | 1.42*** (1.25, 1.62) | 1.35*** (1.18, 1.54) | 1.56*** (1.46, 1.68) |
| Poorer | 2.79*** (1.60, 4.87) | 1.51** (1.17, 1.95) | 2.29*** (1.84, 2.83) | 1.83*** (1.50, 2.23) | 1.70*** (1.47, 1.96) | 1.32** (1.13, 1.54) | 1.71*** (1.58, 1.85) |
| Poorest | 1.83 (0.87, 3.84) | 1.99** (1.36, 2.91) | 2.46*** (1.93, 3.16) | 1.89*** (1.48, 2.41) | 1.58*** (1.34, 1.87) | 1.26* (1.06, 1.51) | 1.70*** (1.55, 1.87) |

(*Continued*)

**Table 3.** (*Continued*)

| Characteristics | 1996 AOR (95% CI) | 1999 AOR (95% CI) | 2004 AOR (95% CI) | 2007 AOR (95% CI) | 2011 AOR (95% CI) | 2014 AOR (95% CI) | Pooled data AOR (95% CI) |
|---|---|---|---|---|---|---|---|
| **Administrative division** | | | | | | | |
| Barisal | | | Ref. | | Ref. | Ref. | Ref. |
| Dhaka | - | - | 0.79 (0.61, 1.02) | - | 1.41*** (1.17, 1.69) | 0.86 (0.72, 1.02) | 1.003 (0.91, 1.11) |
| Chattogram | - | - | 1.12 (0.86, 1.47) | - | 1.53*** (1.24, 1.89) | 0.94 (0.78, 1.13) | 1.24*** (1.11, 1.38) |
| Khulna | - | - | 0.60*** (0.46, 0.79) | - | 1.41*** (1.17, 1.71) | 0.75** (0.63, 0.89) | 0.91 (0.82, 1.01) |
| Rajshahi | - | - | 0.89 (0.67, 1.18) | - | 1.47*** (1.20, 1.80) | 0.80* (0.66, 0.96) | 1.02 (0.92, 1.14) |
| Rangpur | - | - | - | - | 1.24* (1.02, 1.52) | 0.81* (0.67, 0.98) | 1.32*** (1.14, 1.53) |
| Sylhet | - | - | 1.11 (0.81, 1.54) | - | 1.64 *** (1.32, 2.02) | 1.12 (0.90, 1.40) | 1.07 (0.95, 1.20) |
| **Survey year** | | | | | | | |
| 1996 | | | | | | | Ref. |
| 1999 | - | - | - | - | - | - | 0.67*** (0.57, 0.78) |
| 2004 | - | - | - | - | - | - | 0.60*** (0.52, 0.70) |
| 2007 | - | - | - | - | - | - | 0.51*** (0.44, 0.60) |
| 2011 | - | - | - | - | - | - | 0.36*** (0.32, 0.42) |
| 2014 | - | - | - | - | - | - | 0.47*** (0.41, 0.54) |
| Hosmer-Lemeshow p-value | 0.2721 | 0.1053 | 0.0002 | 0.1207 | 0.5444 | 0.2988 | 0.006 |
| AUC | 0.768 | 0.733 | 0.737 | 0.688 | 0.636 | 0.63 | 0.674 |

¶Widowed/Divorced/Not living together,

*p<0.05,

**p<0.01,

***p<0.001,

AOR = Adjusted Odds Ratio, CI = Confidence Interval, AUC = Area Under Curve

women living in the Khulna (AOR: 0.75; 95% CI: 0.63, 0.89), Rajshahi (AOR: 0.80; 95% CI: 0.66, 0.96), and Rangpur division (AOR: 0.81; 95% CI: 0.67, 0.98) in the survey year 2014 had significantly lower likelihoods of having a "low" HIV knowledge.

## Sensitivity analysis

We found similar trends of "low" HIV knowledge from 1996 to 2014 (94.2% to 70.2%) (S1 Fig in S1 File). The overall "low" HIV knowledge was 78%. In the multiple regression model for sensitivity analysis, we found the place of residence, wealth quintile, age at first marriage, and women's level of education were potential correlates of "low" HIV knowledge both in the individual survey and pooled analysis, which is consistent with our main analysis (S3 Table in S1 File).

## Discussion

Due to a lack of proper knowledge regarding HIV transmission and prevention methods, HIV still poses a significant public health concern. The situation is even worse among women who tend to have lower levels of HIV transmission knowledge than men. In this study, we investigated the trends and correlates of "low" HIV knowledge among ever-married women in

Bangladesh using nationally representative BDHS data from 1996 to 2014. We observed an overall decreasing trend of "low" HIV knowledge. However, the prevalence of "low" HIV knowledge was still alarmingly high in 2014. We found that the place of residence, wealth quintile, age at first marriage, and women's level of education were potential correlates of "low" HIV knowledge, which were consistent across every survey period and also in the pooled analysis. Besides, women's age, mass-media exposure, use of condoms during sexual inter-course, and women's marital and employment status were also found significant correlates in specific survey years as well as in pooled analysis.

The percentage of ever-married Bangladeshi women with "low" HIV knowledge decreased from 72% in 1996 to 58% in 2014, indicating an overall improvement, but the prevalence of "low" HIV knowledge still remains high. Similar to our findings, but in the opposite direction, a Vietnamese study found that comprehensive HIV knowledge among ever-married women increased from 26% in 2000 to 42% in 2014 [30]. Similarly, a study across LMICs reported an increase in HIV knowledge among young women (15–24 years) in 24 countries but a decrease in 10 countries between 2003 to 2018 [31].

Our study found that women living in rural areas had significantly higher "low" HIV knowledge than women living in urban areas. Similarly, in our multiple regression model of pooled data and individual survey data (except in the survey year 2011), we explored higher odds of "low" HIV knowledge among rural women than urban women. This finding is consistent with studies conducted in Pakistan, India, Burundi, Ethiopia, Kenya, Sub-Saharan Africa, and other LMICs [31–35]. This discrepancy in HIV knowledge between rural and urban women exists due to the better socio-economic position of urban women [36]. Therefore, policymakers should consider implementing policies to improve women's education, increase access to healthcare facilities, and raise awareness of safe sex practices in rural areas to increase HIV-related knowledge.

In our study, we further observed women in the poorest wealth quintile had higher rates of "low" HIV knowledge than women in the richest wealth quintile across all the study periods. Our multiple regression analysis of pooled and individual survey data found higher odds of "low" HIV knowledge among women who lived in the poorest wealth quintile. Similar to our results, a Nigerian study found low HIV-related knowledge among women of the poorest quintile [37]. The studies conducted in Vietnam, and Malawi found women who belonged to the poorest quintile had low comprehensive HIV knowledge which supports our finding [30, 38]. Further, similar evidence was found in a study conducted across LMICs [31]. The remaining inequalities between women from socio-economic groups highlight the vulnerability of the poorest women. The poorest women, similarly to rural women, have a lack of access to health-care facilities, less participate in health awareness programs, and are not exposed to mass media, which lead to a high proportion of "low" HIV knowledge [39, 40]. To address these issues, policymakers should focus on creating more employment opportunities, providing free education and public health care facilities, facilitating awareness programs on health care, and increasing media exposure for women in the poorest households. These efforts could significantly improve the current lack of HIV knowledge and reduce the vulnerability of women in the poorest wealth quintile.

Our multiple regression model of pooled data further revealed that women who first married at age 18 or older had lower odds of "low" HIV knowledge than those who first married before age of 15 years. This association was consistent across all the separate survey rounds. Although there is no previous literature confirming this association, this finding is important in the context of Bangladesh where the median age at first marriage is low (16 years) [29]. The reason for the lower probability of "low" HIV knowledge among adult women might be due to marrying at a mature age helping women to learn about protected and safe sex and also

assisting them to know about sexually transmitted diseases [41]. It is also expected that after the age of eighteen, women have often completed secondary education, and they gain knowledge during the schooling period. The policymakers should come up with different policies for increasing the age of women's first marriage that not only help to increase women's age at first marriage but also help them to increase HIV knowledge. A consistent mass-media campaign focusing on the adverse effects of early marriage could be a potential manner of increasing women's age at first marriage.

Consistent with the previous studies [17, 30, 31, 42–44], our study explored women's education as a potential correlate of HIV knowledge. Our analysis of both pooled and individual survey data found that women with no formal education and those with a primary level education had significantly higher odds of "low" HIV knowledge than those who completed secondary or higher education. Education is often referred to as a "social vaccine" for preventing the transmission of HIV [45]. Education is the best way for developing knowledge, understanding, and social relations that ultimately help to prevent HIV transmission [45]. Educated mothers are more conscious of the different behavioral strategies including the use of condoms during sexual intercourse, abstinence from sexual intercourse with multiple partners, and exposure to mass media regularly [45]. Policymakers in Bangladesh should prioritize women's education as an integral part of preventing HIV transmission and should design HIV knowledge and awareness programs accordingly.

In our study, in addition to the potential correlates that exist in all survey years, such as age at first marriage, level of education, place of residence, and wealth quintile, we also found that media exposure (except in the survey year 2007) and condom use during sexual intercourse (except in the survey year 1996) were significant correlates of "low" HIV knowledge. Based on these results, we can recommend that the policymakers, programme managers, and programme implementers prioritize certain areas, including rural and poor households, integrating HIV prevention essays into the text curriculum, and promoting education for women, when implementing HIV prevention programmes. This study provides directions for initiating long-term interventions aimed at improving the level of HIV-related knowledge, taking into consideration the identified correlates. Future research is warranted to establish the causal relationship between the level of HIV knowledge and individual, community, and sociodemographic factors.

## Strengths and limitations

This study has some strengths. The data used for the analysis came from nationally representative surveys that followed standardized procedures, which provided a reliable basis for the research. This study depicted trends in HIV knowledge, differential by urban-rural and richest-poorest groups, and identified the most vulnerable groups over nearly two decades. Additionally, the study identified potential correlates of low HIV knowledge that existed throughout the period, which could help to initiate long-term policy interventions. The pooled data analysis also gave indications regarding potential correlates of low HIV knowledge. However, this study had some limitations. The number of questions used to assess knowledge was different across the survey years, although some questions were consistent across all the survey years. Additionally, the study used a sub-sample, which reduced the estimated sample size. Nevertheless, the sample size was large enough to be representative and generalize the findings. Further, this study does not allow us to establish any cause-effect relationships due to the cross-sectional design. However, the cross-sectional study is widely accepted to establish the associations.

## Conclusion

Our study demonstrated that the "low" level of HIV knowledge decreased between 1996 to 2014 among reproductive women, but the proportions of "low" HIV knowledge remained high. We also observed widening disparities in HIV knowledge between women in the urban-rural and richest-poorest groups over time. Age at first marriage, level of education, type of place of residence, and wealth quintile were potential correlates of "low" HIV knowledge across all the survey years and in the pooled estimates. Based on our findings, we recommend including more HIV-related topics in the curricula, promoting social awareness for increasing age at first marriage, disseminating the message about the adverse effect of HIV in the rural area through community leaders, and strengthening existing HIV interventions, particularly targeting the rural areas and the poorest households.

## Supporting information

**S1 File. This file contains all the supporting tables and figure.**
(DOCX)

## Author Contributions

**Conceptualization:** Md. Tariqujjaman, Md. Mehedi Hasan, Rashidul Azad, Md. Arif Hossain, Mahfuzur Rahman, Mohammad Bellal Hossain.

**Data curation:** Md. Tariqujjaman.

**Formal analysis:** Md. Tariqujjaman.

**Investigation:** Md. Mehedi Hasan, Mahfuzur Rahman, Mohammad Bellal Hossain.

**Methodology:** Md. Tariqujjaman, Md. Mehedi Hasan, Mohammad Abdullah Heel Kafi, Md. Alamgir Hossain, Saad A. Khan, Nadia Sultana, Rashidul Azad, Md. Arif Hossain, Mahfuzur Rahman, Mohammad Bellal Hossain.

**Project administration:** Md. Tariqujjaman.

**Software:** Md. Tariqujjaman.

**Supervision:** Md. Mehedi Hasan, Rashidul Azad, Mahfuzur Rahman, Mohammad Bellal Hossain.

**Validation:** Md. Tariqujjaman, Md. Mehedi Hasan, Mohammad Abdullah Heel Kafi, Md. Alamgir Hossain, Saad A. Khan, Rashidul Azad, Md. Arif Hossain, Mahfuzur Rahman, Mohammad Bellal Hossain.

**Visualization:** Md. Tariqujjaman.

**Writing – original draft:** Md. Tariqujjaman, Md. Mehedi Hasan, Mohammad Abdullah Heel Kafi, Md. Alamgir Hossain, Saad A. Khan, Nadia Sultana, Rashidul Azad, Md. Arif Hossain, Mahfuzur Rahman, Mohammad Bellal Hossain.

**Writing – review & editing:** Md. Tariqujjaman, Md. Mehedi Hasan, Mohammad Abdullah Heel Kafi, Md. Alamgir Hossain, Saad A. Khan, Nadia Sultana, Rashidul Azad, Md. Arif Hossain, Mahfuzur Rahman, Mohammad Bellal Hossain.

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
