## [Decision Letter · Decision Letter 0]

11 Jul 2022

PONE-D-21-38006Trends and correlates of low HIV knowledge among ever-married women of reproductive age: Evidence from cross-sectional Bangladesh Demographic and Health Survey 1996-2014PLOS ONE

Dear Dr. Tariqujjaman,

Thank you for submitting your manuscript to PLOS ONE. After careful consideration, we feel that it has merit but does not fully meet PLOS ONE’s publication criteria as it currently stands. Therefore, we invite you to submit a revised version of the manuscript that addresses the points raised during the review process.

Please note that we have only been able to secure a single reviewer to assess your manuscript. We are issuing a decision on your manuscript at this point to prevent further delays in the evaluation of your manuscript. Please be aware that the editor who handles your revised manuscript might find it necessary to invite additional reviewers to assess this work once the revised manuscript is submitted. However, we will aim to proceed on the basis of this single review if possible.  Your manuscript has been assessed by an expert reviewer, whose comments are appended below. The reviewer has highlighted concerns about several aspects of the methodology and rationale for the study. Please ensure you respond to each point carefully in your response to reviewers document, and modify your manuscript accordingly.

We look forward to receiving your revised manuscript.

Kind regards,

Joseph Donlan

Editorial Office

PLOS ONE

Journal Requirements:

https://journals.plos.org/plosone/s/file?id=ba62/PLOSOne_formatting_sample_title_authors_affiliations.pdf,

2. Please include a separate caption for each figure in your manuscript.

Reviewers' comments:

Reviewer's Responses to Questions

**Comments to the Author**

1. Is the manuscript technically sound, and do the data support the conclusions?

Reviewer #1: Partly

2. Has the statistical analysis been performed appropriately and rigorously? 

Reviewer #1: Yes

3. Have the authors made all data underlying the findings in their manuscript fully available?

Reviewer #1: Yes

4. Is the manuscript presented in an intelligible fashion and written in standard English?

Reviewer #1: No

5. Review Comments to the Author

Reviewer #1: Dear Authors,

Thanks for choosing an interesting topic for the manuscript “Trends and correlates of low HIV knowledge among ever-married women of reproductive age: Evidence from cross-sectional Bangladesh Demographic and Health Survey 1996-2014”. I have reviewed your article and have few concerns to improve methodological rigor of your research

Abstract

Line # 47: Within the results, kindly also mention AOR for the year 1996

I would suggest to rephrase the conclusion from lines # 49-51, e.g. instead of ‘reduce’, reduction of disparities between urban-rural and poorest-richest quintiles may address the low level of HIV knowledge …

Introduction

Line # 69: Can you please explain “Rohingya” for understanding of general readers?

I think authors should highlight the prevailing misconceptions and rumors associated with lack of knowledge about HIV.

Line # 93-94: I disagree with authors regarding the given explanation to not considering the ‘ever-heard of HIV’. In DHS, this is an opening question, which applies to only those women, who had ever heard of HIV and excluding those who had not heard HIV. Subsequently, questions related to HIV knowledge are prompted. Authors’ justification is quite vague. I would recommend to highlight the gaps in existing literature, which are being bridged due to this research. Most importantly, how this study is difference from the earlier study (Sheikh, 2017), what value your research is adding into literature? Authors should answer these important questions.

Methodology

Line # 103-104: Here you have taken sample of ever-married women of reproductive age (15-49 years) who ever-heard of HIV. Isn’t it contradictory with your stated research objectives.

Line # 105: Therefore, our sample sizes for this study were 1781, 3660, 7235, 7687 2011, 12512, and 12593 in the survey years … here ‘2011’ seems typo error and must be omitted.

Sampling design: Was that design with 600 clusters (207 urban and 393 rural) remained same throughout varied BDHSs?

Outcome measure:

Authors are advised to highlight why the series of questions for HIV related knowledge is not unform across the varied BDHSs?

Statistical analysis

Kindly mention which statistical software was used for analysis.

I am interested to explore how pooled data was constructed and analyzed

Lastly, I would recommend to add future research directions and implications of this research

6. PLOS authors have the option to publish the peer review history of their article (what does this mean?). If published, this will include your full peer review and any attached files.

Reviewer #1: **Yes: **Dr. Sarosh Iqbal

---

## [Author Response · Author response to Decision Letter 0]

4 Oct 2022

Reviewers' comments:

Reviewer #1: Dear Authors,

Thanks for choosing an interesting topic for the manuscript “Trends and correlates of low HIV knowledge among ever-married women of reproductive age: Evidence from cross-sectional Bangladesh Demographic and Health Survey 1996-2014”. I have reviewed your article and have few concerns to improve methodological rigor of your research

Abstract

Line # 47: Within the results, kindly also mention AOR for the year 1996

I would suggest to rephrase the conclusion from lines # 49-51, e.g. instead of ‘reduce’, reduction of disparities between urban-rural and poorest-richest quintiles may address the low level of HIV knowledge …

Response: We included 1996 as the reference category (AOR =1) and compared it with other categories. As per your suggestion, we have rephrased the conclusion section as-

“The prevention of early marriage, the inclusion of HIV-related topics in the curriculum, reduction of disparities between urban-rural and poorest-richest quintiles may address the “low” level of HIV knowledge among ever-married Bangladeshi women.” (page 2, Lines: 48-50).

Introduction

Line # 69: Can you please explain “Rohingya” for understanding of general readers?

Response: Explained in parenthesis as follows-

 “an ethnic group who mostly follow Islam and resided in Rakhine State, Myanmar” (page 3, Lines: 68-69).

I think authors should highlight the prevailing misconceptions and rumors associated with lack of knowledge about HIV.

Response: We highlighted the prevailing misconceptions and rumors regarding HIV among ever-married women as follows- 

“In addition, many misconceptions and rumors regarding HIV including people get HIV from mosquito bite, sharing foods with a person who has AIDS and by witchcraft or supernatural means are exist among ever-married women” (pages 3-4, Lines: 83-85).

Line # 93-94: I disagree with authors regarding the given explanation to not considering the ‘ever-heard of HIV’. In DHS, this is an opening question, which applies to only those women, who had ever heard of HIV and excluding those who had not heard HIV. Subsequently, questions related to HIV knowledge are prompted. Authors’ justification is quite vague. I would recommend to highlight the gaps in existing literature, which are being bridged due to this research. Most importantly, how this study is difference from the earlier study (Sheikh, 2017), what value your research is adding into literature? Authors should answer these important questions.

Response: Thanks for your concerns. We did not exclude ever-heard of HIV. We analyzed the data for this study based on the sample of ever-married women who ever-heard of HIV. We agree with you that ever-heard of HIV is an opening question in BDHS and the women who responded “yes” subsequently asked the knowledge-related questions (Table 1). 

In the cited paper, they mentioned the knowledge if the women ever-heard of HIV and otherwise no knowledge. But in our study, we calculated the knowledge score based on the individual knowledge question response, which we think makes our study differs from the cited study. 

According to your comment, we also included the women of never-heard of HIV and did the analysis as a sensitivity analysis. We included the results of sensitivity analysis and presented the trends figure and correlates table as supplementary materials. 

Methodology

Line # 103-104: Here you have taken sample of ever-married women of reproductive age (15-49 years) who ever-heard of HIV. Isn’t it contradictory with your stated research objectives.

Response: We mentioned our study objective before the methodology section in the revised version.

“We included ever-married women of reproductive age as our main analysis but in the revised version we included both samples as sensitivity analysis.” (Page 4, Lines: 97-99)

Line # 105: Therefore, our sample sizes for this study were 1781, 3660, 7235, 7687 2011, 12512, and 12593 in the survey years … here ‘2011’ seems typo error and must be omitted.

Response: Thanks. We omitted the typo in the revised version.

Sampling design: Was that design with 600 clusters (207 urban and 393 rural) remained same throughout varied BDHSs?

Response: The number of clusters throughout the BDHSs are not the same across time. The variations in the number of clusters are largely based on the population census used for drawing samples. As the population census was conducted over roughly 10 years interval in Bangladesh, the selected clusters are likely to change after this period. In addition, the number of clusters may change between the successive round of BDHSs as well depending on the sampling strategy and/or selection criteria. 

We have revised as—

“The sampling frame was a complete list of enumeration areas (EAs) or clusters, consisting of either a village, part of a village, or a group of villages. In the first stage of sampling, in 1996, 313 clusters (71 urban and 242 rural), in 1996, 341 clusters (99 urban and 242 rural), in 2004, 361 clusters (122 urban and 239 rural), in 2007, 364 clusters (136 urban and 228 rural), in 2011, 600 clusters (207 urban and 393 rural) and in 2014, 619 clusters (226 urban and 393 rural) were selected throughout Bangladesh using probability proportional to size (PPS) sampling.” (Page) (Page 5, Lines: 123-128).

Outcome measure:

Authors are advised to highlight why the series of questions for HIV related knowledge is not unform across the varied BDHSs?

Response: We acknowledge that the knowledge questions are not uniform across all BDHSs. This is expected as the BDHS questionnaire is upgradable depending on the new knowledge, indicators, aspects and issues generated/identified from up-to-date research findings. We have discussed this in the discussion section. The added sentences are as follows:

“All the HIV-related knowledge questions were not uniform across the surveys because the BDHS questionnaire is upgradable depending on the new knowledge, indicators, aspects, and issues generated/identified from up-to-date research findings.” (Page 5, Lines: 140-142)

Statistical analysis

Kindly mention which statistical software was used for analysis.

Response: We have mentioned the name of the software in the revised version. (Page 8, Lines: 198-199).

I am interested to explore how pooled data was constructed and analyzed

Response: Thank you so much for your concern. The pooled data were constructed by combining all the individual survey data i. e by appending all six surveys datasets into a single dataset.

Lastly, I would recommend to add future research directions and implications of this research

Response: Thanks for your recommendations. We have added the implications and future research directions in the revised version as follows:

“This study would be the directive for initiating long-term interventions for improving the level of HIV-related knowledge considering the identified correlates. Future research is warranted to establish the causal relationship of level of HIV knowledge and individual, community and sociodemographic factors.” (Page 17, Lines: 374-378).

---

## [Decision Letter · Decision Letter 1]

21 Feb 2023

PONE-D-21-38006R1Trends and correlates of low HIV knowledge among ever-married women of reproductive age: Evidence from cross-sectional Bangladesh Demographic and Health Survey 1996-2014PLOS ONE

Dear Dr. Tariqujjaman,

Thank you for submitting your manuscript to PLOS ONE. After careful consideration, we feel that it has merit but does not fully meet PLOS ONE’s publication criteria as it currently stands. Therefore, we invite you to submit a revised version of the manuscript that addresses the points raised during the review process.

The manuscript has been evaluated by two reviewers, and their comments are available below.

The first reviewer is satisfied with the revisions you made to your manuscript, but reviewer two has a number of requests for clarification.

Could you please carefully revise the manuscript to address all comments raised?

We look forward to receiving your revised manuscript.

Kind regards,

Steve Zimmerman, PhD

Associate Editor, PLOS ONE

Journal Requirements:

Reviewers' comments:

Reviewer's Responses to Questions

**Comments to the Author**

1. If the authors have adequately addressed your comments raised in a previous round of review and you feel that this manuscript is now acceptable for publication, you may indicate that here to bypass the “Comments to the Author” section, enter your conflict of interest statement in the “Confidential to Editor” section, and submit your "Accept" recommendation.

Reviewer #1: All comments have been addressed

Reviewer #2: (No Response)

2. Is the manuscript technically sound, and do the data support the conclusions?

Reviewer #1: Yes

Reviewer #2: Yes

3. Has the statistical analysis been performed appropriately and rigorously? 

Reviewer #1: Yes

Reviewer #2: Yes

4. Have the authors made all data underlying the findings in their manuscript fully available?

Reviewer #1: Yes

Reviewer #2: Yes

5. Is the manuscript presented in an intelligible fashion and written in standard English?

Reviewer #1: Yes

Reviewer #2: Yes

6. Review Comments to the Author

Reviewer #1: Dear Authors,

Thanks for sharing the revised version. All comments have been addressed as advised previously. I have no further comments.

Thanks and wish you all the best

Reviewer #2: Dear Authors, Thanks for selecting an interesting and important topic for the manuscript “Trends and correlates of low HIV knowledge among ever-married women of reproductive age: Evidence from cross-sectional Bangladesh Demographic and Health Survey 1996-2014”. I have reviewed the article and have few areas to be improved of your research

Introduction:

Line 61-62: Instead of “recreational drug 62 users (especially those who use needles)” we can say simply “injecting drug users”

Line 65: You wrote Although the prevalence of HIV among the general population in Bangladesh is low. Can you please mention the data/statistics of low for better understanding?

Line 70: Can’t you write: This refugee population is a high-risk group to HIV?

Methodology:

Overall methodology section looks fine.

Sampling design:

Line 128: Please use 30 households were (instead of was) selected

Line 129: systematic random sampling technique

Outcome measure:

Line 149+150: In Table-1, in around 7 rows, the questions included AIDS virus. Should we write AIDS virus, or we would mention AIDS infections and/o illnesses? Please check and if possible, rephrase it though it’s probably picked from BDHS questionnaire.

Covariate measures:

Line 172: What is the elaboration of DHS? If it is first used, the full meaning should mention.

Statistical analysis:

Line 199: We can add StataCorp. LP, College Station, TX, USA instead of College Station, TX, USA only.

Results (Sample characteristics):

Line 167/221-222 (Table 2)/278-279 (Table 3): Check the right spelling of Chattagram/ Chattogram.

Line 69/158/221-222 (Table 2): You mentioned Islam and also Muslim, which one is the right. It should be consistent in the overall manuscript. I would suggest you use: Muslim and Non-Muslim options if there is no reservation.

Trends of HIV-related knowledge from 1996 to 2014:

Line 237: using range, put space before and after (-) like 63.7% - 90% instead of 63.7%-90% for better reading/understanding.

Line 240/245: If you wanted to mean S1 = Supplementary Table 1, in any place, it needs to be defined like S1 = Supplementary Table. Otherwise, it may confuse to the reader.

Discussion:

Line 331: The meaning of “participate in health awareness programs” is not clear. Is it like less participation in health awareness programs?

Line 332: which ultimately causes (or cause?) a high proportion of “low” HIV knowledge….

Line 337: would significantly combat the….. I think, combat is not the right wording here rather we can use to represent the improvement from the currently poor situation.

7. PLOS authors have the option to publish the peer review history of their article (what does this mean?). If published, this will include your full peer review and any attached files.

Reviewer #1: **Yes: **Dr. Sarosh Iqbal

Reviewer #2: **Yes: **Md Rajwanul Haque

---

## [Author Response · Author response to Decision Letter 1]

17 Mar 2023

Reviewer #2: Dear Authors, Thanks for selecting an interesting and important topic for the manuscript “Trends and correlates of low HIV knowledge among ever-married women of reproductive age: Evidence from cross-sectional Bangladesh Demographic and Health Survey 1996-2014”. I have reviewed the article and have few areas to be improved of your research

Response: Thanks a lot for reviewing our manuscript by spending your valuable time. We have tried to revise our manuscript by addressing your valuable comments and suggestions. 

Introduction:

Line 61-62: Instead of “recreational drug 62 users (especially those who use needles)” we can say simply “injecting drug users”

Response: Thanks for your suggestion and direction. We revised accordingly. (page 3; line 61)

Line 65: You wrote Although the prevalence of HIV among the general population in Bangladesh is low. Can you please mention the data/statistics of low for better understanding?

Response: We provided the prevalence in parenthesis. (page 3; line 64)

Line 70: Can’t you write: This refugee population is a high-risk group to HIV?

Response: We dropped this sentence. Thanks.

Methodology:

Overall methodology section looks fine.

Response: Thanks a lot for your compliment.

Sampling design:

Line 128: Please use 30 households were (instead of was) selected

Response: Changed accordingly. (page 5; line 127)

Line 129: systematic random sampling technique

Response: Added random in the revised version. (page 5; line 128)

Outcome measure:

Line 149+150: In Table-1, in around 7 rows, the questions included AIDS virus. Should we write AIDS virus, or we would mention AIDS infections and/o illnesses? Please check and if possible, rephrase it though it’s probably picked from BDHS questionnaire.

Response: Thanks for your valuable comment. We wrote these questions as similar to the BDHS asked to the respondents. We checked again the BDHS questions and found the same. We may keep these to make them consistent with BHDS. 

Covariate measures:

Line 172: What is the elaboration of DHS? If it is first used, the full meaning should mention.

Response: We mentioned the full form of DHS when used in the revised version. (page 8; line 175)

Statistical analysis:

Line 199: We can add StataCorp. LP, College Station, TX, USA instead of College Station, TX, USA only.

Response: Added as per your direction. Thanks. (page 9; line 203)

Results (Sample characteristics):

Line 167/221-222 (Table 2)/278-279 (Table 3): Check the right spelling of Chattagram/ Chattogram.

Response: Corrected Chattagram to Chattogram. (page 8; line 169) (page 10; Table 2) (page 14; Table 3)

Line 69/158/221-222 (Table 2): You mentioned Islam and also Muslim, which one is the right. It should be consistent in the overall manuscript. I would suggest you use: Muslim and Non-Muslim options if there is no reservation.

Response: Thank you so much for pointing out this important issue. We have kept Islam and others including Hinduism, Buddhism, and Christianity as we found these in the BDHS data and also in the BDHS report. It will be consistent to keep the same in both places.

Trends of HIV-related knowledge from 1996 to 2014:

Line 237: using range, put space before and after (-) like 63.7% - 90% instead of 63.7%-90% for better reading/understanding.

Response: Placed accordingly in all the places when we used the ranges. 

Line 240/245: If you wanted to mean S1 = Supplementary Table 1, in any place, it needs to be defined like S1 = Supplementary Table. Otherwise, it may confuse to the reader.

Response: Thanks. Mentioned in the revised version. (page 11; lines 242, 246)

Discussion:

Line 331: The meaning of “participate in health awareness programs” is not clear. Is it like less participation in health awareness programs?

Response: Agreed. We included less in the revised version. (page 16; line 338)

Line 332: which ultimately causes (or cause?) a high proportion of “low” HIV knowledge….

Response: Changed causes to lead. (page 16; line 339)

Line 337: would significantly combat the….. I think, combat is not the right wording here rather we can use to represent the improvement from the currently poor situation.

Response: Agreed. Combat is a vast term. We changed combat to improve. Thanks. (page 16; line 343)

---

## [Decision Letter · Decision Letter 2]

11 May 2023

Trends and correlates of low HIV knowledge among ever-married women of reproductive age: Evidence from cross-sectional Bangladesh Demographic and Health Survey 1996-2014

PONE-D-21-38006R2

Dear Dr. Tariqujjaman

We’re pleased to inform you that your manuscript has been judged scientifically suitable for publication and will be formally accepted for publication once it meets all outstanding technical requirements.

Kind regards,

Mpho Keetile, PhD

Academic Editor

PLOS ONE

Additional Editor Comments (optional):

Reviewers' comments:

Reviewer's Responses to Questions

**Comments to the Author**

1. If the authors have adequately addressed your comments raised in a previous round of review and you feel that this manuscript is now acceptable for publication, you may indicate that here to bypass the “Comments to the Author” section, enter your conflict of interest statement in the “Confidential to Editor” section, and submit your "Accept" recommendation.

Reviewer #2: All comments have been addressed

2. Is the manuscript technically sound, and do the data support the conclusions?

Reviewer #2: Yes

3. Has the statistical analysis been performed appropriately and rigorously? 

Reviewer #2: Yes

4. Have the authors made all data underlying the findings in their manuscript fully available?

Reviewer #2: Yes

5. Is the manuscript presented in an intelligible fashion and written in standard English?

Reviewer #2: Yes

6. Review Comments to the Author

Reviewer #2: I have reviewed the manuscript and found that the authors have addressed all the comments provided by me with complete satisfactory statements. Now the overall manuscript looks fine and there is no comment from my end on the revised version.

7. PLOS authors have the option to publish the peer review history of their article (what does this mean?). If published, this will include your full peer review and any attached files.

Reviewer #2: **Yes: **My name

---

## [Editor Report · Acceptance letter]

17 May 2023

PONE-D-21-38006R2 

Trends and correlates of low HIV knowledge among ever-married women of reproductive age: Evidence from cross-sectional Bangladesh Demographic and Health Survey 1996-2014 

Dear Dr. Tariqujjaman:

I'm pleased to inform you that your manuscript has been deemed suitable for publication in PLOS ONE. Congratulations! Your manuscript is now with our production department. 

Kind regards, 

on behalf of

Dr. Mpho Keetile 

Academic Editor

PLOS ONE